# QSAR Implementation for HIC Retention Time Prediction of mAbs Using Fab Structure: A Comparison between Structural Representations

**DOI:** 10.3390/ijms21218037

**Published:** 2020-10-28

**Authors:** Micael Karlberg, João Victor de Souza, Lanyu Fan, Arathi Kizhedath, Agnieszka K. Bronowska, Jarka Glassey

**Affiliations:** 1School of Engineering, Newcastle University, Newcastle upon Tyne NE1 7RU, UK; karlberg.micael@gmail.com (M.K.); L.Fan5@ncl.ac.uk (L.F.); arathikmenon.2004@gmail.com (A.K.); 2Chemistry—School of Natural and Environmental Sciences, Newcastle University, Newcastle upon Tyne NE1 7RU, UK; J.V.De-Souza-Cunha2@newcastle.ac.uk (J.V.d.S.); agnieszka.bronowska@ncl.ac.uk (A.K.B.)

**Keywords:** monoclonal antibodies, quantitative structure–activity relationship, hydrophobic interaction chromatography, process development, manufacturability, protein dynamics analysis

## Abstract

Monoclonal antibodies (mAbs) constitute a rapidly growing biopharmaceutical sector. However, their growth is impeded by high failure rates originating from failed clinical trials and developability issues in process development. There is, therefore, a growing need for better in silico tools to aid in risk assessment of mAb candidates to promote early-stage screening of potentially problematic mAb candidates. In this study, a quantitative structure–activity relationship (QSAR) modelling workflow was designed for the prediction of hydrophobic interaction chromatography (HIC) retention times of mAbs. Three novel descriptor sets derived from primary sequence, homology modelling, and atomistic molecular dynamics (MD) simulations were developed and assessed to determine the necessary level of structural resolution needed to accurately capture the relationship between mAb structures and HIC retention times. The results showed that descriptors derived from 3D structures obtained after MD simulations were the most suitable for HIC retention time prediction with a R^2^ = 0.63 in an external test set. It was found that when using homology modelling, the resulting 3D structures became biased towards the used structural template. Performing an MD simulation therefore proved to be a necessary post-processing step for the mAb structures in order to relax the structures and allow them to attain a more natural conformation. Based on the results, the proposed workflow in this paper could therefore potentially contribute to aid in risk assessment of mAb candidates in early development.

## 1. Introduction

Monoclonal antibodies (mAbs) have gained increasing popularity over the last three decades in terms of both sales and research investments due to their high specificity and clinical safety. Since the launch of the first therapeutic antibody in 1986, 79 antibodies have been approved by the European Medicine Agency (EMA) and 89 antibodies by the US Food and Drug Administration (FDA) [1,2,3]. In addition, over 550 antibody candidates are currently being reviewed in early clinical trials (Stage I and Stage II) as well as 79 candidates in late-stage development which makes the antibody therapeutics one of the fastest growing segments in the pharmaceutical market [4].

Due to their popularity, biopharmaceutical companies invest several billions of dollars in development for every new mAb candidate, which was estimated to be $2.558 billion on average from start to finish in 2016 [5]. However, more than 90% of all new mAb candidates fail due to undesired effects in clinical trials or being unfeasible for manufacturing. Selection criteria of lead candidates in the preclinical phase are today mainly based on drug specificity, affinity, and potency towards a target antigen, whereas properties related to clinical safety and product quality are generally not thoroughly explored. Consequently, this leads to the selection of candidates with suboptimal properties that might not be feasible for treatment or manufacturing. As Zurdo et al. stated [6], the developability of a mAb candidate can be divided into three subcategories which correspond to pharmacology, clinical safety, and manufacturability, where the selection criteria today are mainly focused on the pharmacology of the candidate. Considering the large investments in the pre-clinical phase alone, which were estimated to $1.098 billion, a robust assessment of an antibodies developability is sorely needed to further characterise antibody properties related to clinical safety and manufacturability in order to decrease failure rates and attrition in biopharma [5,7].

During recent years, multiple high-throughput experimental assays have been designed in order to increase the product knowledge and identify potential problems with the antibody candidates in early development phases [8,9]. A few common experimental assays for characterisation of biophysical properties are hydrophobic interaction chromatography (HIC) which correlates with aggregation propensity [10], cross-interaction chromatography (CIC) which correlates with solubility and poly-specificity [11,12], and affinity-capture self-interaction nanoparticle spectroscopy (AC-SINS) which correlates with viscosity and solubility [13,14]. Though the amount of necessary material needed for high-throughput screening is decreasing with current technologies, accurate developability assessment requires the combination of several screening assays since no single assay appears to be fully predictive. This leads to numerous experiments to be performed which can become cumbersome and time-consuming if many potential candidates need to be assessed in the pre-clinical phase.

It has been suggested that developability assessment should be performed using in silico tools first to assess potential problems with the structure, followed by extended experimental characterisation on promising candidates [15]. Many in silico tools for structural assessment of the mAbs candidates have been developed for evaluation of post translation modifications (PTMs) and degradation pathways based on the candidate sequence that can potentially impact the stability or immunogenicity of the mAb candidate. Examples of such tools include prediction of deamidation or isomerisation of asparagine or glutamine [16], and oxidation of tryptophan and methionine [17], to mention a few. More recently, implementation of quantitative structure–activity relationship (QSAR) for more complex prediction of biophysical properties has become more common. QSAR models are powerful in silico tools which link a measured activity or behaviour (response) to structural properties (descriptors) of a protein, using multivariate data analysis or machine learning approaches [18,19]. The QSAR framework has been extensively applied to develop predictive models to numerous chromatographic applications at different developmental stages such as HIC [20,21,22,23], CIC [24], ion exchange chromatography [25,26], multimodal chromatography [27], and protein A chromatography [28]. The flexibility and the wide applicability of in silico tools for the assessment of mAb developability could therefore be used to highlight potential problems with individual candidates in early development and aid in better selection of lead candidates prior to entering process development [6,29]. This is especially valuable in pre-clinical and early development where little information about the candidate is available.

In this research, a QSAR model was developed for the prediction of HIC retention times (RT), using 81 IgG1-Kappa mAbs from a dataset published by Jain et al. [8]. HIC was explored due to its importance in the assessment of mAb aggregation, solubility, and viscosity, where mAbs with higher HIC RT have been shown to have a negative impact on both clinical safety and manufacturing due to non-specific interactions [10]. Furthermore, an in-depth analysis of the required level of structural resolution for accurate prediction of HIC RT was performed. For this purpose, three unique descriptor sets were developed from three different input sources with increasing structural resolution and fidelity according to:The primary sequence (sequence).Homology model generated from the primary sequence.3D structure obtained after a 50 ns molecular dynamics (MD) simulation using a homology model as a starting point.

The three descriptor sets were named Seq2D, Hom3D, and MD3D, corresponding to the structural representations as listed above. Primary sequence derived descriptors were explored as they allow for fast predictive model development due to ease of implementation where a myriad of different descriptors exists to describe different structural properties. However, due to column binding in HIC being highly dependent on hydrophobic patches on the protein surface [30], it was not possible for descriptors generated from the primary sequence to accurately capture the relationship between structures and HIC RT due to being too simplistic. Meaning that surface properties could not be readily represented when using sequence derived descriptors. Using 3D structures, on the contrary, allowed for more accurate assessment of surface properties and protein stability. Homology modelling and MD simulation were therefore used to generate 3D structures of the antibodies from which structural descriptors were then calculated and linked to the HIC RT. MD simulation was included to alleviate any structural bias that might be present in the generated homology models.

In each structural representation, descriptors were generated from smaller structural components (substructures) of the mAbs corresponding to the complementarity-determining region (CDR) loops and framework regions (FRs) of the variable domains (V_H_ and V_L_), as well as the individual strands of the constant domains in the heavy chain (C_H_1) and light chain (C_L_). Substructures were identified using the international ImMunoGeneTics information system (IMGT) numbering scheme [31,32]. In all structural representations, descriptors were generated using ProtDCal and EMBOSS Pepstat software which are freely available with a friendly graphical user interface and the capacity to generate a high number of molecular descriptors for proteins from FASTA (primary sequence) or PDB (3D structure) files [33,34]. For the primary sequence representation, additional descriptors were generated using amino acid scales. The three resulting descriptor sets were designed to encompass physiochemical, hydrophobic, hydrophilic, topological, and thermodynamic properties of the mAb structures. The relationship between descriptors and HIC RT were captured with ϵ-SVM for regression and evaluated using the root mean square error (RMSE) and coefficient of determination the (R^2^). Descriptor selection was performed on each descriptor set using genetic algorithm with partial least square as a base learner (GA-PLS). Furthermore, the computer aided design of experiment (CADEX) algorithm was used to create an external test set for each descriptor set on which each model’s ability to generalise towards future samples were evaluated.

## 2. Results

### 2.1. HIC RT Prediction from Primary Sequences

When generating the descriptors from the primary amino acid sequences, only the V_H_ and V_L_ domains were considered due to the sequence of the constant domains, C_H_1 and C_L_ being identical between samples. This was an intended modification by Jain et al. who used allele sequences IGHG1*01 and IGLK1*01 for the heavy chain and light chain respectively when expressing all IgG1-Kappa antibodies in their study [8]. Thus, no sequence variability is present in the C_H_1 and C_L_ domains and resulting descriptors would be static in the Seq2D descriptor set. A predictive model was developed on the resulting Seq2D descriptor set following the outlined model development presented in the Materials and Methods section. For a list of descriptors that were calculated for each CDR and FR substructure, please refer to Appendix A.

The resulting predictive model after descriptor selection with GA-PLS showed signs of high model bias (underfitting) where the R^2^ values for the internal cross-validation (model training) were fairly low with a value of 0.56 [35]. This indicated that the model had some difficulty in capturing the correlation between descriptors and the HIC RT responses. In addition, poor generalisation performance in the test set was observed with R^2^ = 0.25. The model predictions using the Seq2D descriptor set are illustrated in Figure 1A where calibration samples (used for model training) and the external test set samples (used for model validation) are depicted in grey and red, respectively.

The high model bias was further assessed by examining the model learning curve (see Figure 1B). The model learning curve (or experience curve) allows for the model performance to be assessed by incrementally increasing the number of samples (experience) and then evaluating the change in RMSE of the calibration, cross-validation, and test set. Fewer samples will yield a lower RMSE in the calibration due to ease of accomplishing a better fit between used samples. However, this will, in turn, result in higher RMSE values in cross-validation due to overfitting and the model will not be able to perform accurate prediction except for the fitted calibration samples. As more samples get added, the RMSE of the calibration will gradually increase due to higher difficulty in fitting a linear line between the increasing number of samples. On the contrary, the RMSE of the cross-validation will decrease with an increased number of samples where the model starts to capture the underlying correlation between descriptors and the HIC RT response. As can be observed, the RMSE of the calibration and the cross-validation approaches plateau values of approximately 0.4 and 0.7, respectively, with an increasing number of samples. Therefore, further addition of samples to increase structural variability in the dataset is unlikely to improve model performance in any significant way [36]. Thus, critical information in the selected descriptors that are correlated to the HIC RT is either missing or confounded by uninformative information. The latter was explored by examining the impact of the mAb species on both the HIC RT and descriptors.

### 2.2. Impact of Species on Primary Sequence Descriptors

A statistical test was performed to evaluate if any difference between species (chimeric, human, and humanised) could be observed in the HIC RT. The non-parametric Kruskal–Wallis test was used instead of one-way ANOVA due to the requirement of normality in the latter method not being fulfilled. Normality was assessed using the Anderson–Darling test where the null hypothesis was rejected in the human and humanised sample groups with *p* = 0.0007 < 0.05 and *p* = 0.005 < 0.05, respectively. The Kruskal–Wallis test showed that no significant difference was present in HIC RT between species (*p* = 0.39 > 0.05), thus indicating that the species of the antibody does not impact the HIC RT.

However, a strong correlation to the species was observed in the generated descriptors when performing classification. CADEX with a stratification scheme was used to retain an 80/20 ratio of the chimeric, humanized, and human samples in the calibration set and test set, respectively [37]. The classification was then performed with C-SVM from the LibSVM toolbox and performance was evaluated with Matthews correlation coefficient (MCC) as well as the class sensitivity and specificity [38]. The MCC metric was used, as a discrete form of Pearson correlation coefficient, and can, therefore, be evaluated in the same way [39,40]. Initial results showed a correlation of MCC = 0.42 in the cross-validation and MCC = 0.71 in the test set, thus indicating a moderate and strong correlation to the species, respectively. Many classification errors in the cross-validation were the result of wrongly classifying the chimeric and human species as humanised with corresponding class sensitivities of 0.29 and 0.53, respectively, where a value of one indicates the correct classification of positive samples. To investigate further, an additional 123 sequences were gathered from the IMGT mAb database in order to increase the number of samples for each species, thus introducing more structural variability in the dataset [41]. A new classification model was developed with the additional sequences which achieved significantly higher discrimination performance of chimeric and human samples with sensitivities of 0.62 and 0.88, respectively, in the cross-validation. This, in turn, yielded a higher correlation between descriptors and species with MCC = 0.73 in the cross-validation and MCC = 0.76 in the test set. Classification performance for both models is presented in Appendix A.

This strongly suggests that the descriptors, developed from the primary sequences, contain information that is highly correlated to the mAb species. This is supported by research that shows that systematic variation of the amino acid composition occurs between different mAb species and is therefore well known [42]. Wold et al. stated that datasets containing systematic variation uncorrelated to response can significantly reduce model performance due to being detrimental [43]. This stems from the fact that many of the Seq2D descriptors are calculated as a sum of tabulated residue values for a specified region e.g., CDR loop or FRs. This means that each residue will impact the final value of each descriptor equally. It is therefore unlikely that the descriptors will contain information that is highly correlated to the HIC RT due to confoundment. This is due to that only a few residues actually contributes in HIC column binding, whereas the majority of the antibody residues does not interact with the hydrophobic ligands of the column [30].

### 2.3. HIC RT Prediction from 3D Homology Models

All-atom homology models were developed for the Fab regions of the 81 IgG1-Kappa samples from the dataset published in Jain et al., using MODELLER. Two of the mAbs: muromonab and teplizumab had to be excluded in this process due to modelling difficulties and poor quality of the models. Therefore, only 79 of the 81 IgG1-Kappa mAbs were used for predictive model development. Individual subsets of descriptors were then generated from the CDR loops and FRs of the variable domains, V_H_ and V_L_. In addition, descriptors were also generated for the constant domains, C_H_1 and C_L_, where each constant domain was divided into smaller fragments that correspond to the seven different strands (A-G) that make up the domain. For both the variable and constant domains, the IMGT numbering scheme was used to identify the start and stop position of each substructure. For a list of descriptors that were calculated for each substructure, refer to Appendix A. The dataset was prepared and split into calibration and test set as described in the material and methods section.

The selected descriptors with GA-PLS showed clear signs of overfitting where the calibration samples were almost perfectly fitted with R^2^ = 0.99 in the calibration, whereas large errors were observed between predicted and experimental measurements in the external test set with R^2^ = −0.08. The negative value of the test set R^2^ indicates that the variation of the prediction error is greater than that of the inherent standard deviation of the measured HIC RT of the test set. The model overfitting can be observed in Figure 2A where calibration samples (grey) fall on the parity line while the test set samples (red) have large errors in their prediction. The model learning curve (see Figure 2B) shows that the error of the test set stabilised after having been trained with 10 or more samples and further additions did not improve performance of the model on the test set. The error of the cross-validation, on the contrary, decreases incrementally which is typical behaviour when predictive models suffer from high model variance. This means that descriptors needed to explain individual sample variation were selected, thus yielding a better model fit for the calibration samples but at the expense of poor performance in the external test set. This, therefore, further supports the observation that GA-PLS struggled to select truly informative descriptors that were correlated to the HIC RT for all antibodies.

Since all homology models were generated based on a single PDB template, it was believed that the generated descriptors might have been biased. As the placement of residues in Cartesian space is decided by pairwise sequence alignment of the template and an antibody sequence of interest, the use of a single template might therefore not accurately represent the 81 IgG1-Kappa antibodies true conformations. Relaxation of the homology models through MD simulation was therefore explored to investigate the impact on the surface descriptors.

### 2.4. Molecular Dynamics Simulation for Protein Structure Relaxation

The generated homology models were used as starting structures in all-atom molecular dynamics (MD) simulations with GROMACS where a single 50 ns production run was performed for each structure. Structural stability of the MD simulations was assessed with the root mean square deviation (RMSD) measured in Angstroms (Å) by comparing the resulting simulated structures at each time frame of the simulation with the initial structure. As shown in Figure 3A, an instantaneous change in the structural conformation occurred at the start of each simulation and reached stable RMSD values, varying between 2–5 Å between mAbs, after roughly 5 ns for most antibodies. The structural change was caused by slight shrinkage of the Fab region, thus resulting in more compact structures (results not shown). The initial change can, therefore, be assumed as relaxation of the polypeptide backbone and sidechains in the antibody structures. This indicates that the homology models represented slightly unfavourable conformations. Three of the antibodies: dinutuximab, eldelumab, and gantenerumab showed additional conformation change during the simulation (see Figure 3B) due to slight movement of the domains in the Fab structure. For dinutuximab and gantenerumab, this change was more gradual and was the result of the C_H_1 and C_L_ domains twisting slightly. The conformational change for eldelumab on the contrary, occurred more rapidly and was caused by the V_H_ domain shifting slightly upwards.

To investigate the relationship between protein dynamics and solvent accessibility, the functionally relevant, essential motions were assessed for each of the trajectories in reference to their average conformation via principal component analysis (PCA). This approach should indicate the absolute magnitude of motions related to the largest principal component. When compared to the average SASA per Fabs, two distinct groups emerged, as showed in Figure 4A. The main group, which is composed of 76 proteins, shows a clear correlation between SASA and the magnitude of motion for the biggest principal motion for the simulation set. The second group consists of three members: dinutuximab, eldelumab, and gantenerumab, which obtained high values for the principal motion magnitude. These high values arise from the shift between the constant and variable domains, discussed in the previous paragraph. Interestingly, the SASA values for the whole protein surface do not directly correlate with the changes in the CDR-H3 region, which is a key motif for antigen binding and a critical subregion for HIC retention. Dinutuximab, eldelumab, and gantenerumab (represented as red dots in Figure 4B) still show stable behaviour for the CDR-H3 SASA in comparison to the rest of the dataset.

Given the fact that from the three outliers, only gantenerumab has its crystal structure solved (PDB: 5CSZ), dynamics for both modelled gantenerumab and its crystal structure were compared. As shown in Figure 5A, the RMSD curve of the modelled gantenerumab is much higher than the one obtained from the simulation of the crystal structure due to the conformational change mentioned previously. In addition, the modelled structure has a higher fluctuation for most of the residue in comparison to the crystal structure (black and red curves in Figure 5B, respectively). The higher flexibility can be explained given the difference between starting structures: the structural alignment between the modelled and the crystal gantenerumab have a backbone RMSD of 8.8 Å between all 230 carbon-α pairs and 1.02 Å for 107 pruned carbon-α atoms (calculated via UCSF Chimera Matchmaker). When the average conformation obtained from the model simulation is compared with the starting crystal structure, the total RMSD decreases to 8 Å, with a RMSD of 1.04 Å for 138 pruned atoms. Both have similar structural composition regarding CDRs and secondary structures; however, the crystal conformation already starts in a configuration that both domains are closer, resulting in a more stable starting conformation. This also indicates that the starting structures are highly sensitive towards the template used, which may hide important structural and biophysical information. MD simulation should overcome this since the obtained conformation after the simulation is closer to the experimental crystal structure, given the lower total RMSD and a higher number of pruned atoms.

The effect of the initial conformational change was explored further by investigating the change in cumulative solvent accessible surface area (SASA) of the residues in the CDR loops. This is due to the fact that CDR loops contain the largest sequence variability in the mAb structure and are directly involved in antigen binding [44]. Therefore, any change in the SASA of the CDR loops would signify a potential change in surface properties such as hydrophobicity and charge that has been shown to be important for HIC binding [45]. It was observed that a small initial change in cumulative SASA of the individual CDR loops occurred in many of the mAbs which then stabilised throughout the rest of the simulations.

This change was quantified for each CDR loop by calculating the absolute difference of SASA averages between simulation intervals 0–5 and 5–50 ns. These intervals were chosen due to the stabilisation of RMSD occurring after roughly 5 ns for most mAbs in the dataset (see Figure 3A). The resulting SASA differences between the first five nanoseconds and the remaining part of the simulation are presented in Figure 6 for each CDR loop. The largest differences were observed in the H3 loop where roughly a third of all mAbs showed a SASA change greater than 50 Å^2^. This is understandable due to the H3 loop having the most diversity in both amino acid composition and length out of all the CDR loops and is considered the largest contributor in antigen binding [46]. The length of the H3 loop in the dataset varied between 6–20 residues, based CDR identification with the IMGT numbering scheme. The remaining CDR loops had a less pronounced change in SASA where more than half of the mAbs varied between 0–25 Å^2^ for the H1, H2, L1, and L3 loops. The SASA of the L2 loop especially only changed slightly in the simulations and varied between 0–25 Å^2^ for all mAbs but one. It was found that the sequence length of the L2 loop was three residues long in the 79 used mAbs from the Jain et al. dataset. Meaning that the change in cumulative SASA for the L2 loop is expected to be much less as compared to the CDR loops H1, H2, L1, and L3 which varied between 5–12 residues in length. 

In addition, a structural comparison was performed using the MatchMaker function in the UCSF Chimera toolbox where the initial homology models were superimposed onto the used template, this was done in order to investigate conformational similarities between the two. Results showed that 44 out of the 79 generated homology structures reported a RMSD of less than two for all possible pruned atoms, meaning that the distance between pairwise aligned backbone carbon-alpha atoms of the template and homology models was less than 2 Å on average. Thirty other molecules had the carbon-alpha RMSD between 3 to 4 Å (Appendix A). With all aspects considered, the generated homology models represented slightly biased conformations due to the used template which in turn led to a misrepresentation of residue SASA values in the CDR loops. The MD simulation, therefore, succeeds in relaxing the structures and the individual mAb structures adopt natural conformations based on their unique amino acid composition and sequence order.

### 2.5. HIC RT Prediction from MD Structures

Surface descriptors were generated from 3D structures taken at the last time frame (*t* = 50 ns) of the MD simulation due to the CDR SASA values remaining stable after the initial conformational change (data not shown). This includes dinutuximab, eldelumab, and gantenerumab which showed additional conformational change during the MD simulation (see Figure 3B). Descriptor generation was performed in the same way as was done for the Hom3D descriptor set.

The observed conformational shift of dinutuximab, eldelumab, and gantenerumab showed in Figure 4A was further explored in order to investigate how the specific motions of the constant and variable domains translated over to the generated SASA descriptors. This to investigate if the three mAbs could be viewed as potential outliers in the data set. To this end, hierarchical clustering analysis (HCA) was used with the farthest neighbour algorithm where pairwise differences between samples were quantified with the Euclidean distance in the descriptor space. In addition, PCA was used to explore the contribution of the generated SASA descriptors to the sample separation between mAbs. The clustering analysis showed that two distinct clusters formed between the majority of mAb samples which are depicted in green and red in Appendix A. However, dinutuximab and eldelumab could not be associated with either cluster due to the high distance between cluster samples and dinutuximab and eldelumab which are depicted in grey in Appendix A. This is further quantified by observing the PCA score plots where eldelumab and dinutuximab are separated by the second and third principal components, shown in Appendix A, respectively. For eldelumab, descriptor contribution to the observed separation originated from the constant domains, C_H_1 and C_L_, by observing the PCA loadings (data not shown) whereas, for dinutuximab, descriptor contribution originated from both the framework regions of the variable domains as well as the constant domains. Based on this information, four individual predictive models were generated with ϵ-SVM for regression to quantify the effect of dinutuximab and eldelumab and/or the constant domain descriptors on the predictive performance according to:Dinutuximab and eldelumab kept in the calibration set and constant domain descriptors kept in the descriptor set.Dinutuximab and eldelumab removed from the calibration set and constant domain descriptors kept in the descriptor set.Dinutuximab and eldelumab kept in the calibration set and constant domain descriptors removed from the descriptor set.Dinutuximab and eldelumab removed from the calibration set and constant domain descriptors removed from the descriptor set.

Model (1) showed relatively high R^2^ values in both cross-validation and a test set of 0.75 and 0.63, respectively, thus showing a reasonable ability to capture the correlation between structure and HIC RT as well as generalising towards future samples. In model (2), the cross-validation performance achieved an R^2^ of 0.80 but the performance of the test set significantly dropped to R^2^ = 0.06, thus losing the ability to generalise towards future samples. This was caused by the removal of eldelumab, which is one of the few mAbs in the data set with a high HIC RT. It is therefore believed that eldelumab acts as a pivot point in the data set which is needed in order to link properties of the mAbs structure to higher HIC RT behaviour. In model (3), both cross-validation and test set performance dropped when removing the constant domain descriptors with resulting R^2^ values of 0.59 and −0.02, respectively. This gives a strong indication of the contribution of the constant domains in column binding in HIC which is corroborated by previously published research [30]. Model (4) showed similar results to model (3) with R^2^ values of 0.57 and −0.02 in the cross-validation and test set, respectively. Based on these findings, model (1) was selected as the final model, with dinutuximab and eldelumab retained in the calibration set as well as the constant domain descriptors retained in the descriptor set.

The model fit is illustrated in Figure 7A in which the calibration samples (grey) and test set samples (red) are depicted. As can be observed, slight overfitting of the calibration samples has occurred whose predictions lie on or close to the parity line, whereas larger errors occur in the prediction of the test set samples. However, the test set showed a clear improvement in performance with R^2^ = 0.63 over the predictive model based on homology structures with R^2^ = −0.08. Based on the errors between predicted and measured HIC RT in the test set, predicted values fall within ±1.32 min of their measured HIC RT based on a 95% student t-test confidence interval.

A diagnostic examination using the model learning curve showed the presence of high model variance as indicated by the incremental decrease of cross-validation error without reaching a stable RMSE value (see Figure 7B). However, the biggest difference between the Hom3D and the MD3D based models is the incremental decrease of the test set RMSE of the latter. This is a strong indication that the error of both the cross-validation and test set can be further improved, which is covered in detail in the discussion.

A Y-Randomisation (or Y-Scrambling) test was used as a final validation step to evaluate the selection of the descriptors [47]. An ϵ-SVM model was trained on a randomised (scrambled) HIC response vector while the sample order in the MD3D descriptor set was kept unchanged. This was repeated 50 times and the average of R^2^ and RMSE for the cross-validation was calculated. A resulting R^2^ value of −2.38 and a RMSE value of 1.72 was obtained. This indicates that no chance correlation occurs in the model and that the selected descriptors are important in describing the relationship between mAb structures and HIC RT responses based on the current dataset.

### 2.6. Structural Descriptors Important for Prediction of HIC RT

A general trend observed in the GA-PLS selected descriptors from the MD3D set was that about 45% of all descriptor belonged to the CDR regions, 31% to the framework regions and the remaining 24% belonged to the strands in both constant domains, C_H_1 and C_L_. This indicates the importance of the structural information contained in the variable domains. This is sensible as the CDRs are the source of the greatest sequence variability in the entire mAb structure which in turn affects surface and thermodynamic properties of both the CDRs as well as framework regions in the variable domains [31]. The effect is not as pronounced in the constant domains due to having identical primary sequence. Instead, the variability present in the MD3D descriptors of the constant domains is likely to be related to conformational differences originating from the molecular dynamics simulations. However, it is important to note that the descriptors from the constant domains should not be disregarded due to dynamic interactions between the constant and variable domains which in turn will affect the generated descriptors [48].

A closer inspection revealed that selected descriptors describing the polar surface areas (${SP}_{polar}{\;\mathrm{and}\;SASA}_{polar})$ and non-polar surface areas (${SP}_{nonpolar}\;\mathrm{and}\;{SASA}_{nonpolar}$) belonged almost exclusively to the CDR regions. Representation of the volume (VOLTAE) and the electrostatic potential (SIEP) generated as part of the TAE descriptors were also commonly found belonging to the CDRs [49]. This is consistent with published research where the CDRs have a pivotal role in binding to the HIC column resin with stronger binding usually occurring when the CDRs are long and hydrophobic [45].

In addition, the stability of the mAb structures played a central role in the prediction of the HIC RT represented mostly by energy-based descriptors. The free energy of conformational entropy, $G_{c}\left(F\right)$, which describes the stability of the protein with regards to the hydrophobic interactions in the protein core was selected in the CDRs, framework regions, and the constant domains. This is supported by published literature where protein stability has been reported to play a pivotal role in HIC binding [50]. Other important energy descriptors of note were the number of estimated water molecules surrounding the surface, W(F), and the entropy energy from the first shell of water, $\Delta G_{w}$, representing the energy contribution from interactions between polar residues and surrounding water molecules. Both these parameters are representative of the solvation/de-solvation energetics of the CDR region, which are characteristics that play a role in environment-CDR interaction energy. Therefore, environment changes, such as excipient composition should affect HIC RT. This is corroborated by salt concentration studies which show that more stable mAbs require a higher concentration of salt to disrupt electrostatic forces and solvation layers on the surface in order to expose hydrophobic patches and promote column binding [30,51].

## 3. Discussion

Early-stage screening of mAbs with in silico tools during the pre-clinical phase, based on the potential to cause developability issues, such as aggregation propensity and solubility problems that arise from nonspecific interactions, would aid in selecting more promising candidates and thereby reduce manufacturing failures and attrition rates. QSAR models can serve as important in silico tools that allow for the prediction of mAb behaviour concerning nonspecific interaction. In this work, a comparison between three descriptors sets (Seq2D, Hom3D, and MD3D) was performed and their applicability evaluated towards the prediction of antibody HIC RT.

### 3.1. Insights from Using Primary Sequences

Based on the results, the primary sequence descriptors used in this research were not suitable for HIC RT prediction due to being unlikely to contain information correlated to the HIC RT response due to confoundment. Although this might explain part of the problem when using descriptors generated from the primary sequence, another important factor to consider is the non-linear relationships between descriptors and the HIC RT response. Hebditch et al. found that the aromatic content and charge of the CDR loops have a non-linear relationship with the HIC RT and that a linear model, therefore, would be unable to capture the underlying correlation [23]. Non-linear methods such as ϵ-SVM with a non-linear kernel or ensemble modelling such as random forest could therefore potentially help in improving model performance. However, the biggest drawback of using non-linear and ensemble methods is that the interpretability of individual descriptors contribution to HIC RT becomes much more difficult to assess [52]. Therefore, it is recommended to use linear models, when applicable, as it allows for much greater interpretability where the effect of descriptors on the HIC RT can be directly assessed. Another factor to consider is that surface properties such as charge and hydrophobicity of the antibodies are not directly represented when using the primary sequence as source for descriptors. Therefore, the selection of descriptors to use as well as the mode of structural representation e.g., full protein, CDR loops etc., becomes challenging when trying to minimise detrimental effects on model performance caused by uncorrelated (or irrelevant) information.

### 3.2. Insights from Using Homology Models

Many of the drawbacks of the primary sequence descriptors are resolved when using 3D models where surface properties and protein stability can be considered more accurately. However, it was shown from the results that many of the resulting homology models were greatly biased towards the used template (PDB: 2FGW), and therefore deviated from their naturally relaxed conformations [53]. This has also been observed in other structure determination software more specialised towards antibodies where Almagro et al. and Teplyakov et al. assessed the predicted conformational accuracy of several canonical structure determination methods [54,55]. The authors showed that the resulting homology models still contained conformational errors when compared to their original crystal structures, which was especially prevalent in the CDR-H3 loop. Therefore, considering that MODELLER was used with a single reference template in this research, the resulting homology models were likely to contain large conformational errors. This in turn would lead to misrepresentation of surface and thermodynamic properties.

This was the main reason atomistic MD simulations were implemented as they allow structures to relax and adopt more natural conformations. In this research, the last time frame was used due to the certainty that most structures had reached stable conformations. However, conformational sensitivity of SASA descriptors is still a concern as it varies slightly from frame to frame due to movement in residue side chains and the protein backbone. Chennamsetty et al. stated that by calculating SASA of residues as simulation averages will represent more naturally occurring SASA values [56]. Future work would therefore be to investigate if simulation averaged SASA values would improve model performance as the protein dynamics are considered in such an approach which otherwise is static at individual timeframes. The second reason for using MD is that additional modifications to the simulation environment can be made. This would include changes made to temperature, pressure, salinity, and addition of co-solvents that can greatly impact the quality and stability of the mAb structures. Osmolality and co-solvents especially are considered important factors that affect many of the so-called critical quality attributes (CQAs) of mAbs in QbD [57]. Each simulation can therefore be adapted to mimic their corresponding experimental setup in order to more accurately investigate different environmental factors effect on the protein structure [58]. However, the biggest drawback with MD is that each simulation is computationally expensive and therefore takes a long time to run.

### 3.3. Considerations and Limitations of the MD3D Based Model

The model based on the MD3D descriptor set achieved higher generalisation ability towards future samples with a R^2^ = 0.63 when compared to similar research performed by Robinson et al. with R^2^ = 0.44, Hebditch and Warwicker with R^2^ = 0.39, as well as Jetha et al. with R^2^ = 0.47 [21,23,27]. This comparison, however, needs to be considered with caution due to differences in samples, descriptors and modelling methods used in respective research. Nonetheless, most exciting is the fact that similar descriptors were found to be important to HIC retention in all instances, thus giving credibility to selected descriptors.

As previously mentioned, further improvement of the MD3D-based model is theoretically possible due to the decreasing RMSE in both the cross-validation and external test set observed in the model learning curve (see Figure 7B). One possibility would be to increase model complexity by using non-linear methods in case that non-linear interactions between descriptors are not captured by a linear model. However, as stated previously, this would decrease the model interpretability which takes precedence from a process development point of view [52]. Another solution recommended by Andrew Ng, would be to add additional samples for model training when available. Due to the decrease in RMSE in the learning curve, additional samples could hypothetically aid in decreasing the RMSE further by allowing for more structural variability to be introduced and considered [36]. In addition, the current dataset suffers from a skewed HIC RT distribution, where most samples have a low measured retention time, whereas only a few have a higher retention time. This is linked to the selection of mAbs used in the study performed by Jain et al., where the authors analysed mAbs that had either been approved or were in clinical Phases II and III at the time of publishing. This effectively reduced the number of mAbs with a higher HIC RT, which are more commonly encountered in the pre-clinical studies and in clinical Phase I. Thus, end-point predictions of mAbs with higher HIC RT are likely to have higher prediction uncertainty due to lack of structural variability that can be linked to such samples. To efficiently circumvent this problem, the dataset should be enriched with samples with high measured HIC RT, preferably those from the pre-clinical phase and clinical Phase I. This would increase both the structural diversity and the retention time range of the dataset. It is crucial to note that when adding new samples to the dataset, the previously selected descriptors from GA-PLS are not likely to be representative of the new structures. In fact, new samples are very likely to fall outside the applicability domain defined by the calibration samples, meaning that the values of their structural descriptors would lie outside of the defined descriptor ranges of the calibration samples [59]. Therefore, reselection of calibration samples to account for added structural variability needs to be performed to increase the applicability domain of the model. This would also necessitate reselection of descriptors with GA-PLS in order to re-evaluate descriptors importance and contribution towards the HIC RT response.

Alternatively, a simplification can be made by converting the model to a classification model where a threshold for the HIC RT is defined to separate well-behaving mAbs from problematics ones. This would allow for the HIC RT to be linked to more general structural motifs instead of relaying on smaller structural differences that would impact the HIC RT. Jain et al. defined a lower limit threshold to be 11.1 min for the HIC RT, which results in a classification MMC value of 0.89 and 0.66 for the calibration and the test set, respectively, when using the model developed in this research (data not shown). Higher values for the test set are however desired with MCC values above 0.80 preferred due to signifying very strong correlation between structural descriptors and HIC RT. This further elucidates that more structural variability and samples with higher HIC RT are necessary to train the model in order to increase model performance and accuracy. From the perspective of drug discovery and early development, it is vital that that samples are correctly predicted/classified in order to be used in risk assessment of mAb candidates due to the high inherent developing costs [5].

Regarding the intrinsic molecular dynamics captured by the MD simulations, it is important to reiterate that 50 ns simulations are unlikely to sample significant conformational changes [60]. Nonetheless, it is a sufficient timescale for local stabilisation and improved sampling. As discussed previously, most of the simulations reached a conformational plateau after 5 ns, but longer simulations should, in principle, increase the variability and accuracy of the model. Nonetheless, within the 50 ns simulation window, we were able to see trajectories that had more significant changes, i.e., gantenerumab. This supports the use of MD simulations on the generated homology models, as MD can greatly improve the 3D models towards a more realistic representation, even in the relatively short timescales used in this work. It is also recommended that several independent simulations are performed for each mAb structure to mitigate the risk of overfitting in predictive model development due to the formation of unique structural conformations. This could aid in generating more representative descriptors due to increased sampling of many different occurring conformations.

Another important aspect to consider is the glycan structure of the antibodies which was not considered in this research. In the data set from Jain et al, all antibodies were expressed as IgG1 using the HEK293 cell line. Therefore, the distribution of glycan structures of each mAb can be assumed to be the same. Meaning that variability in HIC RT will be dependent only on the sequence variability in the variable domains. However, this assumption does not hold true when considering true therapeutic mAbs where the glycan structure distribution will vary from drug to drug. Wada et al. stated that the glycan composition of IgG1 mAbs greatly affected the protein stability and aggregation propensity [61]. Therefore, the proposed model in this research is likely to have a strong bias towards the glycan distribution present in the Jain et al. dataset.

## 4. Materials and Methods

### 4.1. Data Collection

All sequence information, substructure, species, and phase of development for all mAbs were collected from the international ImMunoGeneTics database (IMGT) and the Protein Data Bank (PDB) [41,62]. Sequences were stored in FASTA format and 3D structures in PDB format.

### 4.2. HIC Data

The HIC data for the 81 IgG1-Kappa mAbs were obtained from a previous study performed by Jain et al. where 5 μg IgG samples (1 mg/mL) were spiked in with a mobile phase A solution (1.8 M ammonium sulphate and 0.1 M sodium phosphate at pH 6.5) to achieve a final ammonium sulphate concentration of about 1 M before analysis. A Sepax (Newark, DE, USA) Proteomix HIC butyl-NP5 column was used with a linear gradient of mobile phase A and mobile phase B solution (0.1 M sodium phosphate, pH 6.5) over 20 min at a flow rate of 1 mL/min with UV absorbance monitoring at 280 nm.

### 4.3. Substructure Identification

The CDRs and FRs in the variable domains, V_H_ and V_L_, were identified using the unique IMGT numbering system, which relies on highly conserved amino acids such as Cys23, Trp41, Cys104, and Phe/Trp118 [31]. Similarly, identification of individual strands (A-G) in the constant domains, C_H_1 and C_L_, were also identified by using the IMGT numbering scheme [32]. Each sequence or 3D structure was entered into MATLAB 2016 and, subsequently, user-written algorithms were applied to identify the start and stop positions of the 28 smaller substructures corresponding to the CDRs, FRs and strands based on the IMGT identification rules.

### 4.4. Primary Sequence Descriptors

Descriptors based on the primary sequence were generated using EMBOSS Pepstat and ProtDCal [33,34]. See Appendix A for a list of used EMBOSS Pepstat and ProtDCal descriptors. Smaller sequence fragments were generated corresponding to the individual CDRs and FRs of the variable regions and used directly as input for EMBOSS Pepstat. The full mAb sequence was used as input in ProtDCal due to the software calculating the individual contribution of each residue in the sequence/structure. Descriptors for individual CDRs and FRs were then calculated as the sum of corresponding residue properties from ProtDCal.

In addition, three amino acid scales: Z-scale, T-scale, and MS-WHIM were used to calculate additional physicochemical, topological, and electrostatic sequence properties, respectively [63,64,65]. Each residue was converted into its corresponding amino acid scale values where each component was summed according to the CDR and FR regions identified by IMGT numbering. A list of components for the amino acid scales is presented in Appendix A.

### 4.5. Fab Structure Determination

Fab fragments of the mAbs were prepared for simulation using the available sequences of the variable domains V_H_ and V_L_ provided as supplementary information in the study of Jain et al. The heavy chain was prepared by concatenating the IGHG1*01 (IgG1) sequence corresponding to the C_H_1 domain to the provided V_H_ domains. Similarly, the light chain was prepared by concatenating to the IGLK1*01 (Kappa) sequence to the provided V_L_ domains [66].

Homology models were generated using MODELLER (version 2.17) [67]. In this research, PDB: 2FGW was used as a single template for structure determination [53]. This was done due to the sequence identity between the 2FGW structure and the 81 mAbs was higher than 70% in all instances. Pair-wise cysteines involved in disulphide bridges were restrained where the sulphur atoms were placed at 2 Å from each other in order to properly connect the cysteine residues.

UCSF Chimera (Version 1.13) was used for structure comparison of generated homology models and the template through superimposition with the MatchMaker function [68]. Sequence alignment was performed with BLOSUM-62 per default and the spatial deviation between pairwise aligned alpha carbons was reported in Angstrom (Å). The unpruned values were used in this research, meaning that spatial deviation from non-aligned residues was included in the final values.

### 4.6. MD Simulations

Atomistic simulations of generated Fab homology models were performed with GROMACS (version 5.1.4) using the AMBER99SB-ILDN force field [69]. A cubic solvation box of 10 Å was generated for each Fab structure and solvated using TIP3P water molecules. Next, sodium and chloride ions were added to the system at a concentration of 0.1 M. The bonds were constrained using the LINCS algorithm [70], setting a 2 fs time step. The electrostatic interactions were calculated using the particle-mesh Ewald method [71], with a non-bonded cut-off set at 0.1 nm. All structures were minimised using the steepest descent algorithm for 20,000 of 0.02 nm steps. The minimisation was stopped when the maximum force fell below 1000 kJ/mol/nm using the Verlet cutoff scheme. followed by an NVT equilibration of 20 ps with a time step of 2 fs and position restraints applied to the backbone, a following NPT equilibration (20 ps, 2fs step) with backbone position restraints applied. The temperature was set constant at 300 K by using an alternative Berendsen thermostat (τ = 0.1 ps) [72]. The pressure was kept constant at 1 bar by using a Parrinelo-Rahman barostat with isotropic coupling (τ = 2.0 ps) to a pressure bath [73]. A final production run was performed for 50 ns using the high-performance computing (HPC) service ROCKET at Newcastle University. Atom positions and trajectories of the system were recorded at intervals of 40 ps. All analyses were made using Gromacs software suite. The PCAs were calculated per trajectory, using gmx covar and gmx anaeig routines.

### 4.7. D Structure Descriptors

Like the primary sequence descriptors, the full 3D structure of the Fab region was used as input to ProtDCal on which individual residue contributions were calculated for each descriptor. Descriptors for each substructure i.e., CDR loops, FRs, and strands were then calculated by summing the residue properties corresponding to each region. Used 3D descriptors in this research are listed in Appendix A.

The solvent accessible surface area (*SASA*) for each residue in all Fab structures was calculated using GROMACS, the last time frame at *t* = 50 ns was used for all structures. The relative surface area (*RSA*) for each residue was then calculated according to Equation (1) for each residue:(1)$$RSA_{I}=\;\frac{SASA_{i}}{MaxASA_{i}}$$

The value ${MaxASA}_{i}$ is the theoretical *SASA* maximum for a residue *i* in a Gly-X-Gly conformation which was acquired from research published by Tien et al. in 2013 [74]. The value ${RSA}_{i}$ will therefore lie between zero and one, thus giving surface accessibility in percentages. Surface properties (*SP*) corresponding to hydrophobic and polar patches for a defined region were calculated according to Equations (2) and (3), respectively:(2)$$SP_{Nonpolar}=\sum i\epsilon NPR\frac{SASA_{i}}{MaxASA_{i}}C_{i}^{KD}$$
(3)$$SP_{Polar}=\sum i\epsilon PLR\frac{SASA_{i}}{MaxASA_{i}}C_{i}^{KD}$$
$SP_{Nonpolar}$ is the surface descriptor describing the hydrophobicity, $SP_{polar}$ is the surface descriptor describing the surface polarity, and $C_{i}^{KD}$ is the Kyte-Doolittle value for residue *i*. The *NPR* group corresponds to residues: Ala, Gly, Ile, Leu, Met, Phe, Pro, Trp, and Val, while the PLR group corresponds to residues: Arg, Asn, Asp, Cys, Gln, Glu, His, Lys, Ser, Thr, and Tyr.

### 4.8. QSAR Model Development

Each descriptor set was first curated by removing static descriptors with a standard deviation below 0.0001. Each descriptor set was divided into a calibration (training) and test set via the Kennard–Stone algorithm, maintaining an 80% to 20% split of data [75]. The mAbs with the most structural dissimilarity measured as the largest Euclidean distance between samples in the descriptor space is placed into the training set. This allows for high structural variability to be assessed during model training as well as ensures that no sample in the test set falls outside of the descriptor applicability domain defined by the calibration set [59,76].

Removal of highly collinear descriptors was then performed using the V-WSP algorithm on the calibration set [77]. The descriptor reduction with V-WSP followed the outlined methodology proposed by Kizhedath et al. [29]. Each descriptor set was further optimised through supervised descriptor selection using the GA-PLS algorithm from the PLS Toolbox (Version 8.7) from Eigenvector Research. Due to the chance of models becoming overfitted with a single run of GA, the descriptor selection step was iterated 10 times which allowed for a more diverse population of potential descriptor combinations to be generated. A final descriptor subset was then selected based on the number of times they had been selected in the 10 GA-PLS runs that yielded the lowest cross-validation error, thus mitigating the chance of overfitting by reducing the number of descriptors that increase the fit of individual samples [78]. The GA-PLS parameters were set as follows: population size of 64, maximum generations of 100, mutation rate of 0.005, window width of 1, convergence rate of 50%, 30% initial terms, and crossover set to 2 [79].

The HIC RT were modelled against each descriptor set using the ϵ-SVM algorithm from the LibSVM toolbox with a linear kernel [38]. A grid search approach was used to find the optimal model parameters C and ϵ [80]. The grid points used for C were [10^−5^, 10^−4^, 10^−3^, 10^−2^, 10^−1^, 10^0^, 10^1^, 10^2^, 10^3^, 10^4^], whereas the grid points used for ϵ were [10^−3^, 10^−2.5^, 10^−2^, 10^−1.5^, 10^−1^, 10^−0.5^, 10^0^, 10^0.5^, 10^1^]. This resulted in 90 different parameter permutations that were evaluated in the cross validation. Model training of both the GA-PLS and ϵ-SVM algorithms was performed with repeated random k-fold cross-validation with five splits and 20 iterations [81].

## Figures and Tables

**Figure 1 ijms-21-08037-f001:**
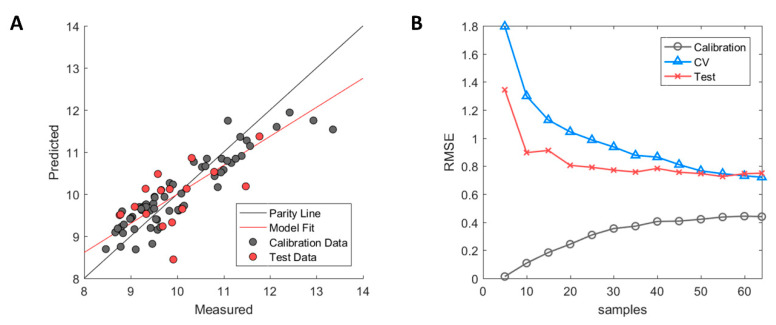
Performance diagnostics of Seq2D based model. (**A**) Measured versus predicted values of calibration (grey) and test set (red) samples. (**B**) Model learning curve showing the change of root mean square error (RMSE) for calibration (grey circle), cross-validation (blue triangle) and test set (red cross). Optimised ϵ-SVM model parameters from the original calibration model were used.

**Figure 2 ijms-21-08037-f002:**
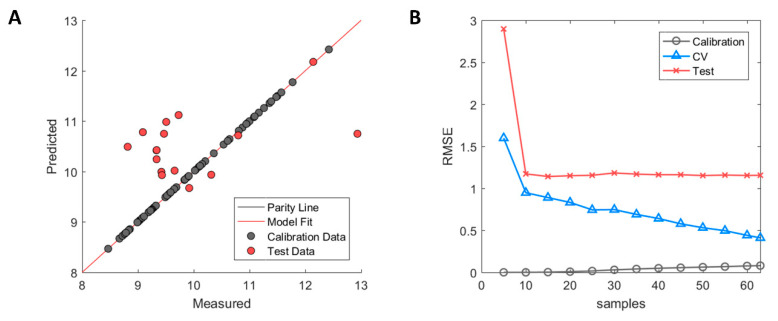
Performance of Hom3D based model after genetic algorithm with partial least square (GA-PLS) selection. (**A**) Measured versus predicted values of calibration (grey) and test set (red) samples. (**B**) Model learning curve for Hom3D based model showing the change of RMSE for calibration (grey circle), cross-validation (blue triangle) and test set (red cross). Optimised ϵ-SVM model parameters from original calibration model were used.

**Figure 3 ijms-21-08037-f003:**
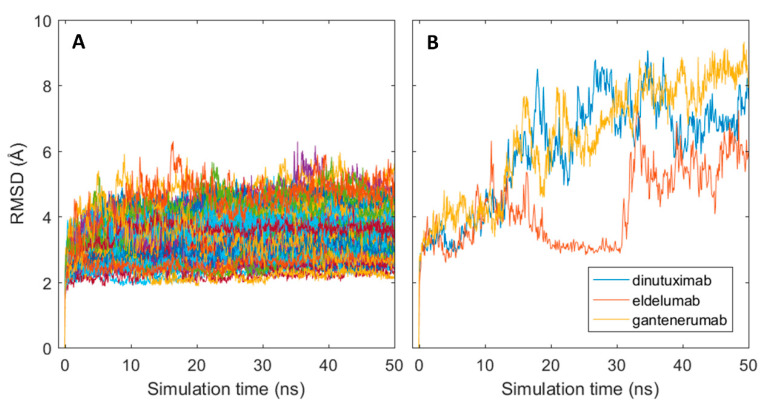
Conformational changes of antibodies from their initial homology models in MD simulations. (**A**) RMSD plots for all simulated mAb; (**B**) RMSD plots for dinuxitumab, eldelumab and gantenerumab.

**Figure 4 ijms-21-08037-f004:**
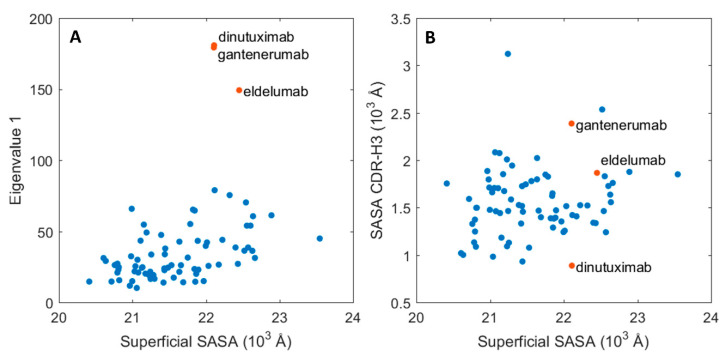
Essential molecular motions of the mAbs and their respective SASA. (**A**) Magnitude of the principal motion in function of the protein superficial SASA. (**B**) CDR-H3 SASA in function of the whole protein SASA. The values for three outliers: dinutuximab, eldelumab, and gantenerumab are depicted in red.

**Figure 5 ijms-21-08037-f005:**
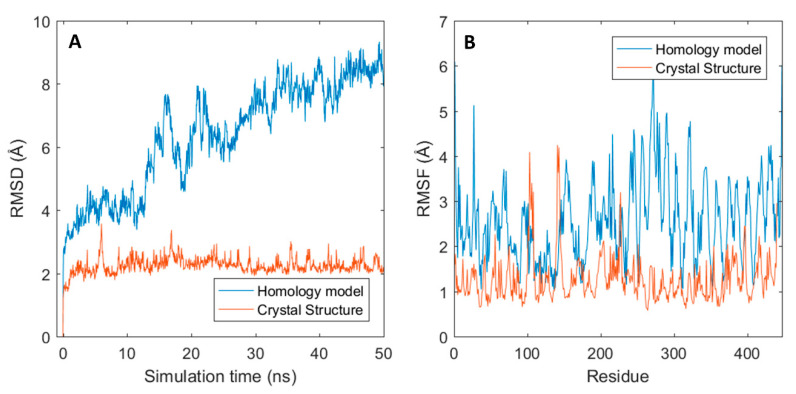
Gantenerumab structural fluctuation for both homology and crystal structure. (**A**) RMSD graphs for both structures through time. (**B**) Root-mean-square fluctuation (RMSF) for all residues.

**Figure 6 ijms-21-08037-f006:**
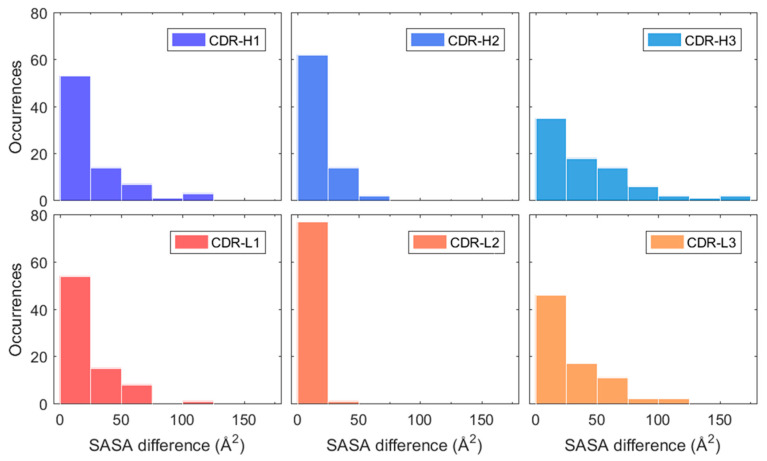
Estimated absolute SASA difference of H1, H2, H3, L1, L2, and L3 between simulation averages of 0–5 and 5–50 ns. The heavy chain CDR loops (H1, H2, and H3) are highlighted in blue and light chain CDR loops (L1, L2, and L3) in red-orange.

**Figure 7 ijms-21-08037-f007:**
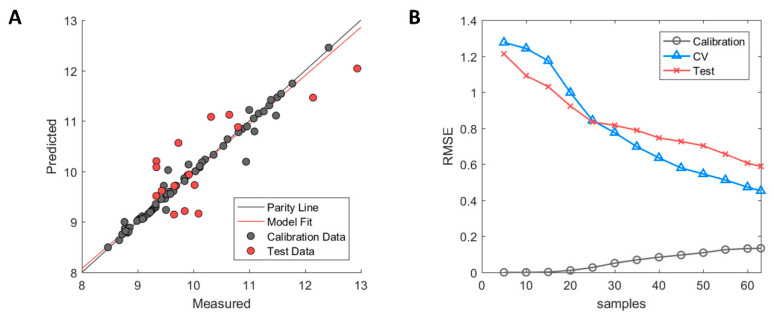
Performance of MD3D based model after GA-PLS selection. (**A**) Measured versus predicted values of calibration (grey) and test set (red) samples. (**B**) Model learning curve for MD-3D based model showing the change of RMSE for calibration (grey circle), cross-validation (blue triangle), and test set (red cross). Optimized ϵ-SVM model parameters from original calibration model were used.

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
