# Peer review of "QSAR Implementation for HIC Retention Time Prediction of mAbs Using Fab Structure: A Comparison between Structural Representations"

_ijms, 2020, doi:10.3390/ijms21218037_

Round 1
Reviewer 1 Report
The manuscript gives a detailed account of research to explore the possibility of predicting Hydrophobic Interaction Chromatography (HIC) retention times of monoclonal antibodies using in silico methods. In general the work is clearly described, the data has been analysed with care and in a critical manner, and discussion and conclusions seem sound and insightful. I have only a few minor comments:
- The overall objective is to create an in silico tool to predict HIC RTs, as an aid to predicting the aggregation properties of potential therapeutic mAbs. Some comment on how accurate such a tool would need to be in order to be of practical use would be worthwhile.
- The authors claim that the reason that the MD-based method gives superior results to the homology-modelling only method is that the dynamics in effect "cures" the homology models of structural inaccuracies - at least in part. But I don't really see any direct evidence to support it. For example in Figure 5 we see the RMSD of Gantenerumab simulations begun from both the homology model and the "true" crystal structure: the homology model clearly is undergoing greater structural change, but is that towards the crystal structure conformation?
- I'm confused by the use of PCA methods (Figure 4) - what is the objective of this analysis? No details of the methodology are provided, as far as I can see. Was PCA run once on the aggregated simulation data, or individually on each simulation? If the former, how were the disparate structures/trajectories integrated? If the latter, what is the significance of the analysis as there is no guarantee that the eigenvectors are the same, so a comparison of magnitudes of corresponding first eigenvalues is fairly meaningless.
Author Response
Dear Ms. Marina Ollé Hurtado,
We would like to thank the reviewers for assessing the revised version our manuscript (ID: ijms-968232) and for providing some excellent comments, which we have addressed. All the corrections suggested by both reviewers were addressed in the revised manuscript.
Responses and clarifications to specific comments by reviewer 1 are given below:
Comment 1: The overall objective is to create an in silico tool to predict HIC RTs, as an aid to predicting the aggregation properties of potential therapeutic mAbs. Some comment on how accurate such a tool would need to be in order to be of practical use would be worthwhile.
Authors response: This is an excellent point. For the model to be useful we would argue that the performance of calibration, cross-validation and test set should have an R2 above 0.80 in order to decrease the difference between measured and predicted values. We have expanded the discussion of the MD3D based model to further strengthen this point but also discussed the importance of additional data for model training, especially addition of samples with higher HIC RT (highlighted sections, pp.14-15).
Comment 2: The authors claim that the reason that the MD-based method gives superior results to the homology-modelling only method is that the dynamics in effect "cures" the homology models of structural inaccuracies - at least in part. But I don't really see any direct evidence to support it. For example, in Figure 5 we see the RMSD of Gantenerumab simulations begun from both the homology model and the "true" crystal structure: the homology model clearly is undergoing greater structural change, but is that towards the crystal structure conformation?
Authors response: This is correct. MD simulation of model gantenerumab resulted in an ensemble which is structurally closer to the crystal conformation. The MD simulations were able to reduce the total RMSD between the average conformation from the model simulation and the starting crystal structure from 8.8 Å to 8 Å, while increasing the total aligned (pruned) backbone carbon-α atoms from 107 to 138. However, the RMSD value still is high, since the variable and constant domains are closer in the crystal structure than they are in the model (highlighted sections, pp.8). This difference should be reduced with longer simulations, which we aim to do in the future.
Comment 3: I'm confused by the use of PCA methods (Figure 4) - what is the objective of this analysis? No details of the methodology are provided, as far as I can see. Was PCA run once on the aggregated simulation data, or individually on each simulation? If the former, how were the disparate structures/trajectories integrated? If the latter, what is the significance of the analysis as there is no guarantee that the eigenvectors are the same, so a comparison of magnitudes of corresponding first eigenvalues is fairly meaningless.
Authors Response: The PCA was applied to each individual simulation. Therefore, the diagonalization and resulting eigenvectors are unique per simulation. We agree that these eigenvectors may not show similar motions between different simulations, however, their eigenvalues represent the magnitude of important motions specific to each simulation. On which, this brings information that some systems require larger motions (regardless of their direction) to obtain a stable conformation, even with a comparable value of SASA (highlighted sections, pp.7).
We hope that our responses thoroughly address all the concerns regarding the results, data analysis and interpretation, and we are looking forward to the feedback.
Yours sincerely,
Prof. Jarka Glassey
Reviewer 2 Report
The manuscript titled “QSAR Implementation for HIC Retention Time Prediction of mAbs Using Fab Structure: A Comparison between Structural Representations” proposes a new SVM-based QSAR model to predict HIC Retention Time Prediction of mAbs. The accompanying finding of the study is that MD simulation of mAbs significantly improves the accuracy of its modelled 3D structure as compared to reference crystal structure. The molecular modelling in this study is thorough and utilizes well-known methods. Each step of the modelling procedure is comprehensively discussed.
The overall purpose of the study is to develop the way to predict HIC retention time of mAbs and in such a way to identify those mAb candidates that fail developability criteria. However, the data set used to derive the QSAR model consists of successful mAbs in their late stages of development only. Thus, the majority of mAb candidates that fail developability criteria will fall outside the applicability domain of the QSAR model. It is not clear how the model developed with successful structures can generate accurate predictions useful to detect structures that will fail during the development stage.
Overall, the manuscript is written clearly and has a distinct logical structure. It deserves to be published after addressing the issue mentioned above.
Author Response
Dear Ms. Marina Ollé Hurtado,
We would like to thank the reviewers for assessing the revised version our manuscript (ID: ijms-968232) and for providing some excellent comments, which we have addressed. All the corrections suggested by both reviewers were addressed in the revised manuscript.
Responses and clarifications to specific comments by reviewer 2 are given below:
Comment: The overall purpose of the study is to develop the way to predict HIC retention time of mAbs and in such a way to identify those mAb candidates that fail developability criteria. However, the data set used to derive the QSAR model consists of successful mAbs in their late stages of development only. Thus, the majority of mAb candidates that fail developability criteria will fall outside the applicability domain of the QSAR model. It is not clear how the model developed with successful structures can generate accurate predictions useful to detect structures that will fail during the development stage.
Authors response: This is an excellent point and we have expanded the discussion of the MD3D based model regarding developability (highlighted sections, pp.14-15). As stated, the authors, Jain et al., only included phase II, phase III and approved mAbs in their dataset which effectively limits the number of samples with higher HIC RT which are commonly found in the pre-clinical phase and phase I. We therefore recommend that additional data should be included for model training, preferably with higher HIC RT. This in order to increase the model applicability domain but also to better link structural properties and motifs to that of higher retention times to decrease prediction uncertainty in this region.
We hope that our responses thoroughly address all the concerns regarding the results, data analysis and interpretation, and we are looking forward to the feedback.
Yours sincerely,
Prof. Jarka Glassey